# Multifaceted Determinants of Sexual Intercourse with Non-Regular Female Sex Partners and Female Sex Workers among Male Factory Workers in China—A Cross-Sectional Survey

**DOI:** 10.3390/ijerph192316008

**Published:** 2022-11-30

**Authors:** Kechun Zhang, Siyu Chen, Shiben Zhu, Yuan Fang, Huachun Zou, Yong Cai, Bolin Cao, He Cao, Yaqi Chen, Tian Hu, Zixin Wang

**Affiliations:** 1Longhua District Center for Disease Control and Prevention, Shenzhen 518110, China; 2Centre for Health Behaviors Research, Jockey Club School of Public Health and Primary Care, The Chinese University of Hong Kong, Hong Kong, China; 3Department of Health and Physical Education, The Education University of Hong Kong, Hong Kong, China; 4School of Public Health (Shenzhen), Sun Yat-sen University, Shenzhen 518107, China; 5Kirby Institute, University of New South Wales, Sydney, NSW 2052, Australia; 6School of Public Health, Shanghai Jiaotong University School of Medicine, Shanghai 200025, China; 7School of Media and Communication, Shenzhen University, Shenzhen 518060, China

**Keywords:** non-regular female sex partners, female sex workers, male factory workers, determinants, socioecological model, cross-sectional study, China

## Abstract

With a stratified multi-stage sampling approach, 1361 male factory workers in the Longhua district of the Shenzhen Municipality of China were selected to investigate the multifaceted determinants of sexual intercourse with non-regular female sex partners (NRP) and female sex workers (FSW) among them. The results showed that 24.5% and 21.2% of participants had sexual intercourse with NRP and FSW in the past 6 months, respectively. More specifically, at the individual level, perceived higher job stress and maladaptive coping styles were linked with a higher likelihood of having sexual intercourse with NRP and FSW (adjusted odds ratios [AOR] ranged from 1.06 to 1.17). At the interpersonal level, those who had higher exposure to information related to sexual intercourse with NRP or FSW were more likely to have sex with these female sex partners (AOR: 1.08 & 1.11). At the social structural level, perceived social norms supporting multiple sex partnerships were linked with a higher likelihood of having sexual intercourse with NRP and FSW (AOR: 1.10 & 1.11). No interaction effects were found between the variables at different levels. Providing pre-employment training to clarify roles and job duties, introducing adaptive coping strategies, and addressing misconceptions of social norms are useful strategies to reduce sexual intercourse with NRP or FSW.

## 1. Introduction

Human immunodeficiency virus (HIV) and sexually transmitted infections (STI) are global public health challenges. Worldwide, there were 1.5 million new HIV infections and 7.1 million–82 million new STI cases in 2020 [1,2]. In China, the HIV and STI burden has been increasing over the last few decades. According to the China National Bureau of Statistics, the prevalence of HIV and STIs increased from 33.09 cases per 100,000 residents in 2009 to 51.91 cases per 100,000 residents in 2019, with an annual increase of 4.61% [3]. Heterosexual contacts continue to be the major mode of HIV and STI transmission in China; 69.9% of HIV infections and 30.6–70% of STIs were attributed to heterosexual contacts in 2020 [4,5]. 

Male factory workers were the target population in this study. Most of the factory workers in Shenzhen are internal migrants [6]. In recent years, the driving force behind heterosexual HIV transmission in China has been commercial heterosexual contact and non-marital, non-commercial heterosexual contact [7,8]. The majority of non-marital heterosexual transmissions occur in males [8]. Furthermore, rural-to-urban male migrants had a higher likelihood of engaging in non-marital heterosexual behaviors than males who are non-migrants [9,10]. Among 2925 new HIV cases reported in Shenzhen between 2016 and 2018, 79.5% were male, and 83.7% were migrants [11]. In the literature, the HIV prevalence among male factory workers ranged from 8.5% in Ethiopia [12] to 42.7% in Lesotho [13]. In India, 41.7% of male factory workers reported at least one STI symptom [14]. A high prevalence of either HIV or STI among male factory workers was observed in China [15,16,17]. Although the HIV prevalence among male factory workers in China (0.1%) was lower than in these countries, it was much higher than that of the general population (e.g., 0.3% in students) [15]. Another study indicated that approximately 20–30% of male factory workers in China reported some STI symptoms [16]. Sexual intercourse with high-risk female sex partners (e.g., non-regular female sex partners [NRP] or female sex workers [FSW]) was prevalent among male factory workers in China (over 40% in the past year). Such behaviors are driven by forces of HIV and STI in this group [17,18]. 

It is important to understand factors associated with sexual intercourse with these high-risk female sex partners among male factory workers to inform effective health promotion and service planning. Previous studies suggested that the majority of factory workers are young and away from home [6,17]. Many experienced social isolation, psychological distress, or other barriers to social integration [19]. To avoid social exclusion, they might engage in hazardous sexual behavior or seek commercial sex under peer pressure [19]. These factors were considered in this study. There was a dearth of studies investigating factors of sexual intercourse with these high-risk female sex partners among male factory workers in China.

The socioecological model (SEM) was used as the conceptual framework for this study [20,21], as the model considered determinants of health behavior at different levels, including individual-level, interpersonal-level, and social-structural-level. The SEM is commonly used to understand determinants of HIV-related risk behaviors among the high-risk population across countries [20,22]. The model treats the interaction between factors at different levels with equal importance to the influence of factors within a single level. At the individual level, job stress is high among factory workers in China. Common sources of job stress include workload, role conflict (having different and incompatible roles at the same time), role ambiguity (insufficient information to perform their jobs adequately or when performance evaluation methods are unclear), and underutilization of skills [6]. Job stress was associated with an increased number of sexual partners [23], a greater likelihood of either HIV or STI occurrence [24,25], and decreased condom use [26]. Psychological adaptation involves the emotions or behaviors that people use to deal with stressors to safeguard their psychological health [27]. Maladaptive emotions and behaviors (e.g., depression, anxiety, drinking, and substance use) were well documented among male factory workers due to the lack of social support in China [28], which was associated with higher sexual risk behaviors [19]. In addition to psychological adaptation, coping styles would affect sexual behavior among male factory workers. Coping styles refer to cognitive and behavioral approaches to minimize stress occurrences [29]. Previous studies showed that coping styles could be considered in two dimensions; adaptive coping (positive efforts to overcome stress) and maladaptive coping (negative efforts to avoid and deny stress) [30,31]. Adaptive coping styles (e.g., problem-solving efforts) were protective factors, such as consistent condom use [32]. However, maladaptive coping styles (e.g., substance use, denying, self-distracting, and venting feelings) were associated with higher sexual risk behaviors [32]. The hypothesis was that higher job stress, poorer psychological adaptation, and maladaptive coping styles would be associated with a higher likelihood of having sex with NRP or FSW among male factory workers in China. 

At the interpersonal level, the use of the internet and social media may be a risk factor in sexual intercourse with NRP and FSW. The Social Learning Theory suggests that social behaviors are learned by observing and imitating peers’ behaviors [33]. Observing peers’ sexual behaviors with NRP and FSW may affect one’s attitudes and behaviors. Such observations may occur online and offline. Location-aware social media platforms popular among Chinese society (e.g., WeChat, Weibo, Tantan, and TikTok) have facilitated sexual intercourse with NRP and FSW in one’s geographical location. Similar studies found that young men’s high-risk sexual behavior with NRP was related to their frequent exposure to online pornography through the internet and social media [34,35,36]. 

At the socio-structural level, perceived social norms may influence sexual intercourse with NRP and FSW among male factory workers. Social norms refer to informal rules that reflect a specific behavior (e.g., sexual intercourse with NRP and FSW) in groups and societies [31]. Perceived social norms supporting sexual behavior with NRP and FSW may be a facilitator for male factory workers to regulate and intimate. Prior studies showed that perceived social norms had considerably facilitated risky sexual behavior among colleges and adolescents in the United States [37,38,39]. Furthermore, perceived social support is potentially important as a protective factor of psychological health promotion [40]. Perceived social support refers to how people view their friends, family members, and others as sources of psychological and practical support when needed [41]. A study in Canada found that higher perceived social support was associated with a lower likelihood of engaging in risky sexual behavior among men who have sex with men [42]. No study has applied SEM to identify the related factors associated with sexual intercourse with NRP and FSW among male factory workers in China. 

To address the knowledge gap, this study has the following research questions:(1)What is the prevalence of sexual intercourse with NRP and FSW among male factory workers in China?(2)Are factors at the individual level (job stress, psychological adaptation, and coping styles), interpersonal level (exposure to social media and perceived social support), and socio-structural level (perceived social norms) determinants of sexual intercourse with NRP and FSW among male factory workers in China?

## 2. Materials and Methods

### 2.1. Study Design

This is a secondary analysis of a cross-sectional study among factory workers in Shenzhen between October and December 2019 [6]. Shenzhen Municipality, bordering Hong Kong SAR to the south, is a major special economic zone in China. The majority of the factories here are located in the Longhua district of Shenzhen. There were 1517 factories and more than 1 million factory workers in 2020 [43].

### 2.2. Participants and Data Collection

The original cross-sectional survey participants were full-time factory employees in Shenzhen aged 18 years or above. Details of the sampling methods were described in published papers [6]. A stratified multi-stage sampling approach was used. In Shenzhen, the majority of the factories are located in Longhua. Names of the factories listed in the most up-to-date registry kept by the Longhua CDC (about 2000) were input into an excel file. Using the function of “select random cells,” 16 factories were randomly selected. These factories included 4 machinery processing plants, 3 electronic equipment manufacturers, 3 printing and dyeing plants, 2 chemical raw materials factories, 1 smelter, 1 clothing factory, 1 food and beverage factory, and 1 other factory. Three to four workshops were then randomly selected from each factory. All eligible factory workers in the selected workshops were invited to participate in the study. Interested participants paid a visit to the Longhua District Center for Disease Control and Prevention (CDC). On-site, trained field workers briefed prospective participants about the study and confirmed their eligibility. The fieldworkers guaranteed anonymity and participants’ right to withdraw from the study at any time, and refusals would have no consequences. With written informed consent, participants completed a self-administered questionnaire that took about 30 min to complete. Upon completion, a cash coupon of ¥20 (US$2.6) was given to the participants as a token of appreciation. Out of 2700 workers approached, 2023 completed the survey. The response rate was 75%. This study focused on heterosexual male factory workers. Therefore, participants who were female (*n* = 662) and those who self-reported ever having had oral or anal sex with men (*n* = 0) were excluded; 1361 participants were included in the analysis. There were no missing values in this study. Ethics approval was obtained from the ethics committees of the School of Public Health, Sun Yat-sen University (2019/3).

### 2.3. Measure

#### 2.3.1. Background Characteristics of the Participants

Participants reported sociodemographic information, including age, ethnicity, relationship status, education level, monthly personal income, whether they were living with partners or children, and whether they were frontline workers or management staff. The utilization of HIV testing and other either HIV or STI prevention services in the past 6 months was also measured. 

#### 2.3.2. Sexual Behaviors with NRP and FSW

An FSW was defined as a woman who exchanged sex for money or gifts, and an NRP was defined as a woman who was neither an FSW nor their regular female sex partner (stable girlfriend or wife). The same definitions have been used in many published studies targeting heterosexual males in China [44,45,46,47,48]. The following questions were used to measure their sexual behaviors with NRPs and FSWs: (1) did you have sexual intercourse with an NRP in the past 6 months? (2) If yes, did you have condomless sex with an NRP in the past 6 months? (3) Did you have sexual intercourse with an FSW in the past 6 months? And (4) If yes, did you have condomless sex with an FSW in the past 6 months? 

#### 2.3.3. Individual-Level Variables

In this study, job stress, psychological adaptation, and coping styles were considered individual-level variables. We used the 13-item Job Stress Questionnaire to measure job stress [49]. This scale was validated in the Chinese population [50]. The Job Stress Questionnaire consists of 4 subscales measuring workload, role conflict, role ambiguity, and utilization of skills (response categories: from 1 = “never” to 7 = “always”). Higher scores indicated perceived higher job stress. The Workload Subscale consisted of 5 items with a total score ranging from 5 to 35, and the McDonald’s omega was 0.90. The Role Conflict Subscale consisted of 2 items with a total score ranging from 2 to 14, and the McDonald’s omega was 0.82. The Role Ambiguity Subscale consisted of 3 items with a total score ranging from 3 to 21, and the McDonald’s omega is 0.69. The Utilization of Skills Subscale consisted of 3 items with a total score ranging from 3 to 21, and McDonald’s omega is 0.84.

Psychological adaptation was measured by the Brief Psychological Adaptation Scale [45]. The items were adapted to the context of Shenzhen. Sample items included “Excited about being in Shenzhen.” Items were rated on a 7-point Likert scale (from 1 = “never” to 7 = “always”). The total score on this scale ranged from 8 to 56. Higher scores mean higher levels of psychological adaptation [6,51]. The McDonald’s omega for this scale was 0.81. 

Four subscales of the Coping Orientation to Problems Experienced Inventory (Brief COPE inventory) measuring avoidant coping styles were used [52]. The Brief COPE inventory was validated in the Chinese population [53] and measured coping styles, including substance use, denial, self-distraction, and venting. Participants were asked to rate how often they used these coping styles to deal with stress on a 4-point Likert Scale (from 1 = “I have not been doing this at all” to 4 = “I have been doing this a lot”). The total score of these subscales ranged from 2 to 8. The McDonald’s omega was 0.73 for the Substance Use Subscale, 0.65 for the Denial Subscale, 0.71 for the Self-distraction Subscale, and 0.63 for the Venting Subscale.

#### 2.3.4. Interpersonal-Level Variables

The influence of social media and perceived social support were considered interpersonal-level variables. The validated Influence of Social Media Scale [33] was adapted to measure the frequency of exposure to information related to sexual behavior with NRPs or FSWs on social media platforms popular among Chinese society (WeChat, QQ, Weibo, TikTok, etc.) in the past 6 months. The original scale measured the influence of social media on psychoactive substance use in the Chinese population [33]. In this study, the phrase “psychoactive substance use” was replaced with “sex with NRPs or FSWs,” and 2 items about exposure to online pornography were added to the original scale based on the feedback provided by factory workers. The response categories for these measurements were 1 = “almost never”, 2 = “seldom”, 3 = “sometimes”, and 4 = “always”. Higher scores indicated a higher level of exposure to information from social media. The total score on this scale ranged from 5 to 15. The McDonald’s omega of this scale was 0.87 in this study. The validated Chinese version of the Perceived Social Support Scale was used to measure perceived emotional and instrumental support from family members and friends, with the response categories ranging from 0 (“not at all”) to 10 (“very much”) [54]. The total score on this scale ranged from 0 to 40. The McDonald’s omega of the scale was 0.82 in this study.

#### 2.3.5. Socio-Structural-Level Variables

Perceived social norm was considered a social-structural level variable. Three items measured perceptions of social norms related to multiple sex partnerships. These items were adapted from those measuring perceived social norms related to casual sex among ethnic minorities in China [31], including: (1) factory workers commonly believe that having more than 1 sex partner at the same time is acceptable, (2) factory workers believe that it is acceptable for people with regular sex partners to have other sex partners at the same time, and (3) factory workers think the more sex partners, the better. The Social Norm Scale was formed by summing up individual item scores, with a higher score indicating perceptions of social norms supporting multiple sex partnerships. The total score was between 3 and 15. The McDonald’s omega was 0.85. 

### 2.4. Statistical Analytic Strategy 

SPSS version 26.0 (IBM Crop., Armonk, NY, USA) software was used for statistical analysis. First, descriptive statistics of all categorical variables at each level were computed and presented with mean and standard deviation (SD). Second, item parceling (summing up all items together) and confirmatory factor analysis techniques were used to calculate the value of mean and standard deviation and the reliability coefficient of each subscale, respectively. Third, multiple logistic-regression analysis involving 1 independent variable of interest (e.g., individual- and interpersonal-level factors) and all significant background characteristics (*p* < 0.05); adjusted odds ratios (AOR) and respective 95% confidence intervals (CI) were used to explore the determinants of sexual intercourse with non-regular female sex partners (NRP) and female sex workers (FSW) among them by using the enter estimation method to estimate the parameters. Fourth, multicollinearity analysis for the independent variables in each logistic regression model was investigated by Variance Inflation Factor (VIF) and tolerance [55]. Linearity in the logit was investigated using the Box–Tidwell Test. The existence of outliers was identified by Casewise Diagnostics, and a 3-standard deviation default was used in the study. Hosmer and Lemeshow goodness-of-fit tests were calculated to assess the strength of the multivariable models used in this study [56]. Overdispersion was investigated using studentized permutations. Moreover, interaction terms were created by multiplying 2 independent variables at different levels. To test the significance of the interaction terms, multiple logistic regression models, each adjusted for significant background characteristics, were fit and contained 2 main effect variables and their interaction terms. 

## 3. Results

### 3.1. Background Characteristics of the Participants

The majority of participants were 18–30 years old (50.8%), currently single (58.5%), Han ethnicity (84.9%), without permanent residency in Shenzhen (98.5%) or tertiary education (93.7%), not living with either a partner or spouse in Shenzhen (70.8%), with a monthly personal income lower than ¥5000 (US$1564.9; 70.5%) and working as frontline workers (81.4%). In the past 6 months, 6% and 21.7% of them received HIV testing and STI testing, respectively (Table 1). 

### 3.2. Sexual Behaviors with NRP and FSW

In the past 6 months, 24.5% and 21.2% of the participants had sexual intercourse with NRPs and FSWs, respectively. The prevalence of condomless sex was 21.3% during sexual intercourse with NRPs, and 13.5% during sexual intercourse with FSWs (Table 1).

### 3.3. Variables at the Individual, Interpersonal and Socio-Structural Levels

About 10% of the participants were either sometimes or always exposed to sharing personal experiences supporting sex with either NRPs or FSWs and invitations made by acquaintances or strangers to have sex with them through social media. About 15% of them had unintended exposure to online pornography. The mean scores and SDs of four subscales of the Job Stress Scale, the Brief Psychological Adaption Scale, four subscales of the Brief COPE inventory, the Perceived Social Support Scale, and the Perceived Social Norm Scale are shown in Table 2. The total variance extracted by one factor is 14.8% by Harman’s one-factor test, and it is less than the recommended threshold of 50% [57].

### 3.4. Factors Associated with Sexual Intercourse with NRP

Older age, married to a woman, and living with either a partner or children were associated with a lower likelihood of sexual intercourse with NRPs in the past 6 months (Table 3). After adjusting for these background characteristics, participants who perceived higher role conflict (AOR: 1.06, 95% CI: 1.02, 1.11, Cohen’s d: 0.02) and higher role ambiguity (AOR: 1.04, 95% CI: 1.01, 1.07), and used denial coping styles (AOR: 1.13, 95% CI: 1.03, 1.23, Cohen’s d: 0.003), were more likely to have sexual intercourse with NRPs in the past 6 months. At the interpersonal level, higher exposure to information related to sexual intercourse with either NRPs or FSWs from social media (AOR: 1.08, 95% CI: 1.04, 1.13, Cohen’s d: 0.02) and perceived social norms supporting multiple sex partnerships (AOR: 1.11, 95% CI: 1.06, 1.16, Cohen’s d: 0.05) were also positively associated with sexual intercourse with NRPs (Table 4). The multivariable models did not indicate a multicollinearity issue (tolerance = 0.5 and VIF = 1.0) and had an acceptable fit (Hosmer and Lemeshow test ranged from 0.05 to 0.32). 

### 3.5. Factors Associated with Sexual Intercourse with FSW

Participants who were married to a woman and older were less likely to have sexual intercourse with FSWs in the past 6 months (Table 3). After adjusting for these two variables, perceived higher role conflict (AOR: 1.06, 95% CI: 1.01, 1.10, Cohen’s d: 0.02), using maladaptive coping styles such as substance use (AOR: 1.13, 95% CI: 1.02, 1.26, Cohen’s d: 0.06), and denial (AOR: 1.17, 95% CI: 1.07, 1.28) were associated with a higher likelihood of sexual intercourse with FSWs in the past 6 months. In addition, higher exposure to information related to sexual intercourse with either NRPs or FSWs from social media (AOR: 1.11, 95% CI: 1.07, 1.16, Cohen’s d: 0.04) and perceived social norms supporting multiple sex partnerships (AOR: 1.10, 95% CI: 1.05, 1.15, Cohen’s d: 0.03) were positively associated with sexual intercourse with FSWs (Table 4). The multivariable models did not indicate a multicollinearity issue (tolerance = 0.5 and VIF = 1.0) and had an acceptable fit (Hosmer and Lemeshow test ranged from 0.05 to 0.80). 

### 3.6. Interactions between Independent Variables at Different Levels

None of the interaction terms was significant (Appendix A). There were no interaction effects between variables at different levels. 

## 4. Discussion

This is one of the first studies examining the prevalence of sexual intercourse with NRP and FSW and its associated factors among male factory workers in China. Factors at the individual level, interpersonal level, and social-structural level were associated with sexual intercourse with NRP or FSW. Using the SEM was helpful in understanding sexual risk behaviors in this group from a more comprehensive perspective. The findings provided a knowledge basis to develop tailored behavioral interventions to reduce the risk of HIV and STI for this group. 

The prevalence of sexual intercourse with NRP (24.5%) and FSW (21.2%) in this study was much higher than that reported among the general population (7.4% with NRPs and 4.2% with FSWs) in China [17,58]. Such prevalence was also higher than that observed among male factory workers in other countries (e.g., 3.7% of young migrant factory workers in Nepal reported sex with FSWs) [18]. However, the prevalence of condomless sex with NRPs (21.3%) and FSWs (13.5%) was lower than that reported among general male adults in Thailand (54% with NRP and 39% with FSW) [59]. Despite their high-risk profiles, the utilization of HIV testing (6%) and STI testing (21.7%) was very low. It is possible that some workers with HIV or STI do not know about their infection status. Interventions to reduce sexual risk behaviors and the promotion of either HIV or STI testing are needed for this group. 

The findings provided some empirical implications for developing behavioral interventions tailored to male factory workers. Similar to previous studies, being younger and single were associated with a higher likelihood of having sexual intercourse with NRP and FSW [17,60]. This group may be sexually active and do not have stable girlfriends, which makes them meet their sexual needs by seeking non-regular partners or commercial sex [17]. Future programs should give more attention to male factory workers who are younger and single. 

In line with the hypothesis, a positive correlation was found between job stress and sexual intercourse with NRPs and FSWs in male factory workers. Previous studies found job stress was a significant risk factor for either HIV or STI among miners and truck drivers [24,25]. It is possible that seeking NRPs and FSWs was considered a strategy to cope with job stress by male factory workers. Health authorities and employers should be aware of the potential negative influences of job stress on sexual health and either HIV or STI transmission. Employers should consider providing sufficient information on roles, duties, and evaluation mechanisms during pre-employment training to reduce job conflicts and ambiguity. In line with previous studies, maladaptive coping styles were positively associated with sexual intercourse with NRPs and FSWs [23,32]. Thus, facilitating male factory workers to apply adaptive coping strategies may be useful in reducing sexual risk behaviors among male factory workers. Future programs can introduce some adaptive strategies that are effective in coping with job stress, such as engagement with social gatherings with friends [61], seeking emotional support from families, friends, or psychological professionals [62], taking rests [63], and doing physical exercise [64]. 

At the interpersonal level, higher exposure to information related to sexual behavior with NRPs and FSWs from social media was also significantly associated with sexual intercourse with NRPs or FSWs. This is consistent with previous studies among university students [33,65,66]. The Social Learning Theory explains how social media may influence male factory workers’ sexual behavior with NRPs and FSWs [33]. Some male factory workers may perceive sexual behavior with NRPs or FSWs as normative, as many social media users may express approval for these behaviors in their posts [67,68]. Social networking apps can provide a quick way to connect with NRPs or FSWs nearby [34,35,36]. In addition, receiving a personal invitation to engage in sexual behavior with NRPs or FSWs through these apps may make male factory workers less likely to refuse. Given the strong influence of social media, future programs should consider using these channels to deliver healthy communication messages reducing sexual risk behaviors. 

At the socio-structural level, perceived social norms supporting multiple sex partnerships were associated with a greater likelihood of sexual intercourse with NRP and FSW. Such findings were similar to those observed in previous studies targeting textile factory workers and Yi minority residents [13,37]. According to the Social Norms Theory, misconceptions about how one’s peers think and act influence health-related behavior [39]. The theory indicates that addressing misconceptions of perceived norms will either diminish problem behavior or promote desired behavior [39]. Health authorities may establish a unique platform to deliver norm-based interventions to reduce multiple sex partnerships. Additionally, creating programs that impose regulations banning multiple sex partnerships may create norms discouraging multiple sex partnerships among male factory workers. 

This study had some limitations. First, information from factory workers who refused to join this study was unavailable, and selection bias might exist. Second, only male factory workers in one city were included in this study. Generalizations should be applied cautiously to other Chinese cities. Third, self-reported responses might have reported bias despite the anonymous study. Participants may respond in socially desirable ways to underestimate the prevalence of sexual intercourse with NRPs and FSWs. Fourth, this was a secondary analysis that was not supported by sample size planning. Moreover, a previous study suggested that sum scoring was often contrasted with factor analysis as a competing method. Some researchers used factor analysis to validate the scale but subsequently summed the score of the scale, which employed a model that was different from the validation model. The validated scales used in this study contained one principal factor in the factor analysis. Last but not least, this was a cross-sectional study that could not establish causal relationships.

## 5. Conclusions

In conclusion, about one-quarter of male factory workers in Shenzhen reported sexual intercourse with either NRPs or FSWs. Such behaviors might be a driving force of HIV and STI transmission in this group. Providing pre-employment training to clarify roles and job duties, introducing adaptive coping strategies, and addressing misconceptions of social norms related to multiple sex partnerships may be useful strategies in future programs. These programs should also consider using social media channels to deliver health communication messages.

## Figures and Tables

**Table 1 ijerph-19-16008-t001:** Background characteristics and sexual behaviors of the participants (*n* = 1361).

Characteristics	*n*	%
**Sociodemographic**		
Age (years)		
18–30	692	50.9
31–40	487	35.8
41–50	146	10.7
>50	36	2.6
Ethnicity		
Han	1155	84.9
Minority	206	15.1
Registered permanent residents of Shenzhen		
No	1341	98.5
Yes	20	1.5
Relationship status		
Currently single	660	48.5
Having a stable girlfriend	145	10.7
Married to a woman	556	40.9
Education level		
Junior high or below	774	56.9
Senior high or equivalent	501	36.8
Either college or university and above	86	6.3
Living with either partner or spouse or children in Shenzhen		
No	964	70.8
Yes	397	29.2
Monthly personal income, ¥ (US$)		
<3000 (469.5)	168	12.3
3000–4999 (469.5–782.3)	792	58.2
5000–9999 (782.9–1564.8)	393	28.9
≥10,000 (1564.9)	8	0.6
Status as frontline workers or management staff		
Frontline workers	1108	81.4
Management staff	253	18.6
**Either HIV or STI prevention service utilization in the past 6 months**		
Use of HIV testing		
No	1279	94.0
Yes	82	6.0
Use of other either HIV or STI prevention services (receiving free) condoms and pamphlets and attending workshops/seminars)		
No	1066	78.3
Yes	295	21.7
**Sexual behaviors in the past 6 months**		
Sexual intercourse with non-regular female sex partners (NRP)		
No	1028	75.5
Yes	333	24.5
Sexual intercourse with female sex workers (FSW)		
No	1072	78.8
Yes	289	21.2
Condomless sex with NRP (among those who had sexual intercourse with NRP in the past 6 months, *n* = 333)		
No	262	78.7
Yes	71	21.3
Condomless sex with FSW (among those who had sexual intercourse with FSW in the past 6 months, *n* = 289)		
No	250	86.5
Yes	39	13.5

**Table 2 ijerph-19-16008-t002:** Item responses and scale scores of variables at individual and interpersonal levels (*n* = 1361).

Factors	N or Mean	% or SD
**Individual-level factors**		
Job stress		
Score of the Workload Subscale	15.4	7.9
Score of the Role Conflict Subscale	4.4	2.9
Score of the Role Ambiguity Subscale	8.0	4.0
Score of the Utilization of Skills Subscale	9.3	4.9
Score of the Brief Psychological Adaption Scale	36.9	8.3
Coping styles		
Substance Use Subscale	2.7	1.2
Denial Subscale	3.2	1.4
Self-distraction Subscale	4.1	1.8
Venting Subscale	3.8	1.5
**Interpersonal-level factors**		
Frequency of exposure to the following content on the Internet and social media platforms in the past 6 months, *n* (%) either sometimes or always		
Sharing of personal experiences supporting sex with NRP or FSW	126	9.3
Someone you know inviting you to have sex with them through Internet or social media	145	10.7
Some strangers invite you to have sex with them through the internet or social media	140	10.3
Actively searching for online pornography	198	14.5
Unintended exposure to online pornography (i.e., open a link or a website containing pornography when you are searching other topics)	212	15.6
Influence of Social Media Scale	2.5	3.1
Perceived social support		
Level of emotional support from family members	6.9	3.2
Level of emotional support from either friends or colleagues	6.5	3.3
Level of instrumental support from family members	6.1	3.2
Level of instrumental support from either friends or colleagues	5.5	3.1
Perceived Social Support Scale	25.0	10.4
**Socio-structural-level variables**		
Perceived social norms related to multiple sex partnerships agree/strongly agree		
Factory workers commonly believe that having more than one sex partner at the same time is acceptable	101	7.4
Factory workers believe that it is acceptable for people with regular sex partners to have other sex partners at the same time	72	5.3
Factory workers think the more sex partners, the better	78	5.7
Perceived Social Norm Scale	5.7	2.8

**Table 3 ijerph-19-16008-t003:** Associations between background characteristics and sexual intercourse with non-regular female sex partners and female sex workers in the past 6 months (*n* = 1361).

	Sexual Intercourse with Non-Regular Female Sex Partners		Sexual Intercourse with Female Sex Workers	
	OR (95% CI)	*p* Value	OR (95% CI)	*p* Value
**Sociodemographic**				
Age (years)				
18–30	Reference		Reference	
31–40	1.18 (0.91, 1.53)	0.22	1.14 (0.86, 1.50)	0.36
41–50	0.37 (0.21, 0.63)	<0.001	0.41 (0.23, 0.72)	0.002
>50	0.48 (0.18, 1.25)	0.13	0.44 (0.16, 1.28)	0.13
Ethnicity				
Han	Reference		Reference	
Minority	1.37 (0.98, 1.90)	0.06	1.23 (0.87, 1.74)	0.25
Registered permanent residents of Shenzhen				
No	Reference		Reference	
Yes	1.33 (0.51, 3.49)	0.56	1.24 (0.45, 3.44)	0.68
Relationship status				
Currently single	Reference		Reference	
Having a stable girlfriend	0.86 (0.57, 1.29)	0.47	0.98 (0.64, 1.49)	0.92
Married to a woman	0.54 (0.41, 0.71)	<0.001	0.61 (0.46, 0.81)	0.001
Education level				
Junior high or below	Reference		Reference	
Senior high or equivalent	1.01 (0.78, 1.32)	0.92	0.78 (0.59, 1.03)	0.08
Either College or university and above	1.07 (0.64, 1.79)	0.79	0.89 (0.52, 1.54)	0.68
Living with either partner or spouse or children in Shenzhen				
No	Reference		Reference	
Yes	0.72 (0.55, 0.96)	0.03	0.76 (0.57, 1.03)	0.07
Monthly personal income, ¥ (US$)				
<3000 (469.5)	Reference		Reference	
3000–4999 (469.5–782.3)	1.08 (0.73, 1.60)	0.70	1.09 (0.73, 1.63)	0.69
5000–9999 (782.9–1564.8)	0.94 (0.61, 1.44)	0.76	0.78 (0.50, 1.23)	0.28
≥10,000 (1564.9)	3.20 (0.77, 13.38)	0.11	2.20 (0.50, 9.65)	0.30
Status as frontline workers or management staff				
Frontline workers	Reference		Reference	
Management staff	1.14 (0.84, 1.56)	0.41	1.07 (0.77, 1.49)	0.70
**Either HIV or STI prevention service utilization in the past 6 months**				
Use of HIV testing				
No	Reference		Reference	
Yes	1.47 (0.91, 2.38)	0.12	1.49 (0.90, 2.45)	0.12
Use of other either HIV or STI prevention services (receiving free) condoms and pamphlets and attending either workshops or seminars)				
No	Reference		Reference	
Yes	1.17 (0.87, 1.57)	0.30	1.29 (0.96, 1.75)	0.10

OR > 1 means a higher score in either item or scale was associated with a higher likelihood of having the outcome (e.g., more likely to have sexual intercourse with NRPs or FSWs). OR < 1 means a higher score in either item or scale was associated with a lower likelihood of having the outcome (e.g., less likely to have sexual intercourse with NRPs or FSWs).

**Table 4 ijerph-19-16008-t004:** Factors associated with sexual intercourse with non-regular female sex partners and female sex workers in the past 6 months (*n* = 1361).

	Sexual Intercourse with Non-Regular Female Sex Partners			Sexual Intercourse with Female Sex Workers		
	AOR (95% CI)	*p*-Value	Cohen’s d	AOR (95% CI)	*p*-Value	Cohen’s d
**Individual-level factors**						
Job stress						
Workload Subscale	1.02 (1.00, 1.03)	0.07	0.001	1.01 (0.99, 1.02)	0.44	0.001
Role Conflict Subscale	1.06 (1.02, 1.11)	0.04	0.02	1.06 (1.01, 1.10)	0.01	0.02
Role Ambiguity Subscale	1.04 (1.01, 1.07)	0.02	0.003	1.02 (0.99, 1.06)	0.18	0.001
Utilization of Skills	1.02 (1.00, 1.05)	0.09	0.01	1.01 (0.98, 1.04)	0.55	0.002
Brief Psychological Adaption Scale	1.00 (0.99, 1.02)	0.99	0.01	1.00 (0.99, 1.02)	0.91	0.001
Coping styles						
Substance Use Subscale	1.10 (1.00, 1.22)	0.06	0.004	1.13 (1.02, 1.26)	0.02	0.06
Denial Subscale	1.13 (1.03, 1.23)	0.01	0.06	1.17 (1.07, 1.28)	0.001	0.08
Self-distraction Subscale	0.98 (0.92, 1.06)	0.64	−0.06	1.01 (0.94, 1.09)	0.78	0.01
Venting Subscale	1.06 (0.98, 1.15)	0.16	0.02	1.06 (0.97, 1.15)	0.19	0.02
**Interpersonal-level factors**						
Influence of Social Media Scale	1.08 (1.04, 1.13)	<0.001	0.02	1.11 (1.07, 1.16)	<0.001	0.04
Perceived Social Support Scale	0.99 (0.98, 1.00)	0.20	−0.003	0.99 (0.98, 1.00)	0.05	−0.01
**Socio-structural-level variables**						
Perceived Social Norm Scale	1.11 (1.06, 1.16)	<0.001	0.05	1.10 (1.05, 1.15)	<0.001	0.03

OR > 1 means a higher score in either item or scale was associated with a higher likelihood of having the outcome (e.g., more likely to have sexual intercourse with NRPs or FSWs). OR < 1 means a higher score in either item or scale was with a lower likelihood of having the outcome (e.g., less likely to have sexual intercourse with NRPs or FSWs). AOR: adjusted odds ratios, odds ratios adjusted for significant background characteristics listed in Table 3.

## Data Availability

The data presented in this study are available from the corresponding author upon request. The data are not publicly available as they contain personal behaviors.

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
