# Peer review of "Multifaceted Determinants of Sexual Intercourse with Non-Regular Female Sex Partners and Female Sex Workers among Male Factory Workers in China—A Cross-Sectional Survey"

_ijerph, 2022, doi:10.3390/ijerph192316008_

Round 1

Reviewer 1 Report

Comments:

Abstract: “Sexual intercourse with non-regular female sex workers (NRP) and female sex workers 23 (FSW) may be risk factors of HIV/STI transmission among factory workers.”

This sentence should be revised.

“Sexual intercourse with female sex workers may be risk factors of HIV/STI transmission among factory workers”

What was the sampling method?

Introduction:

Male factory workers are vulnerable to HIV and STIs.  Why? Please cite a reference.

A high prevalence of HIV/STI among male factory workers was observed in China. (Ref??)

Methods:

This is a secondary analysis of a cross-sectional study among factory workers in Shenzhen between October and November 2019 (13).

In mentioned study “Relationships Between Job Stress, Psychological Adaptation and Internet Gaming Disorder Among Migrant Factory Workers in China: The Mediation Role of Negative Affective States

authors have said “This cross-sectional study ….between October and December 2019” Please revise the date of conduction of the study.

“ First, 16 factories were randomly selected by the research team” by which methods were factories selected?

Methods, line 142: “On-site, trained fieldworkers 142 briefed prospective participants about the study and confirmed their eligibility”

What was the eligibility of participants? I checked your first article in that too I still do not see anything about the eligibility of participants.

Well-trained fieldworkers briefed prospective participants about the study and confirmed their eligibility to participate in the study”

2.3.2 Sexual behaviors with NRP and FSW

“Participants were asked whether they had sexual intercourse with NRP or FSW in the past six months. Among participants who had such experiences, we further asked whether they had condomless sex with these types of female sex partners”.

Authors asked just these two questions for sexual behaviors of participants?

Please under the titles of “Individual-level variables, Interpersonal-level variable, Social-structural-level variables” before explaining the questionnaires briefly name what aspect of Individual-level variables, Interpersonal-level variable, Social-structural-level variables were considered in this study.

For instance, in this study “X”, “Y” were considered as Individual-level variables. After briefly descriptive, explain the questionnaires, score, validity,,,,,, of every variables If abbreviations are used in the text they should be defined in the text at first use. Brief COPE???

Higher score indicated perceived higher job stress. What does mean higher score? 20? 50? 70? 100?

Max- and Min/cutoff (if applicable) of all questionnaires used in this study should be mentioned.

One factor was identified by exploratory factor analysis, explaining for 66.5% of total variance.

Was exploratory factor analysis done in this study? 

One factor was identified by exploratory factor analysis. Which was that factor? Then that factor was deleted? Or that factor rested in the study?

About 15% of them actively searched for online pornography or exposed to online pornography without active searching. What was the definition actively here? In method section authors talked about “The response categories for these measurements were 1=almost never, 2=seldom, 3=sometimes, 4=always”

Actively in result section means always? If yes, please try to use the same word.

Score of the Workload Subscale 1 (mean, SD) was 15.4. I can understand that score was good? Bad? Because authors did not write about the min-max questionnaires/ scales, subscales…

Result section is so difficult to follow. Unfortunately there are a lot variables, content in this study which is not possible to be understood or interpreted.

“Factors at the individual level, interpersonal level, and social-structural level were determinants of sexual intercourse with NRP or FSW” this sentence is not correct. Because you authors did not explore them by qualitative study. From the beginning of your study, they applied those questionnaires to measure the level of association.

“This study had some limitations. First, we did not collect information from participants who refused to participate in the study” this sentence is not correct. 

Reviewer 2 Report

Reviewer’s Comments

October 28, 2022

*Issues on the Introduction Section

1-The authors mentioned the “socioecological model” in line 71 and used it as the conceptual framework for the present study. However, its description in the manuscript is too simple and it missed at least one key part of this model― interaction of the different levels of the model. The authors should stress the ecological framework treats the interaction between factors at the different levels with equal importance to the influence of factors within a single level. Please do this work.

2-The authors used “Multilevel determinants of sexual intercourse with non-regular female sex partners and female sex workers among male factory workers in China―a cross-sectional survey” as the title for the study. The word “multilevel” is really confusing and it gives readers a feeling that the models you used were multilevel logistic regression models. However, what the authors did are just regular logistic regression analysis. I suggested the authors change it “multifaced determinants”.

*Issues on the Methods Section

3-On line 128, the authors mentioned “Shenzhen”. In fact, it should be “Shenzhen Municipality”.

Similarly, Hongkong should be “Hongkong SAR” (Special administrative region of PRC).  For people without background knowledge of China, they do not know the meaning of them.

4-Below the subtitle “2.2. Participants and data collection”, the authors described the sampling method and specific procedure. However, there is not any introduction of missing values in the sample. What is the missing value rate? What is the missing value pattern? Please make sure if the missing data is MCAR (Missing completely at random) for each variable in the sample.

5-In line 154, “measurement” should be “measure”.

6-From line 168 to line 224, please remove all Cronbach’s a value. Because the authors do not understand what does a mean. the authors don’t fully understand how this coefficient is obtained and when it can be used. Frankly, Cronbach's alpha coefficient, also known as tau-equivalent reliability, is a reliability coefficient that provides a method of measuring internal consistency of tests. The assumption of tau-equivalence (i.e., the same true score for all test items, or equal factor loadings of all items in a factorial model) is a requirement for α to be equivalent to the reliability coefficient. Tau-equivalence is a strong assumption, one that isn’t typically evaluated in practice. If the authors forced to compute α coefficient when there are no correlated measurement errors, it will underestimate the real reliability. Please compute “omega” coefficient instead for each scale.

7-Have you ever checked the Harman’s one-factor test to make sure there is no common variance method bias?

8- Please explain why the authors performed exploratory factor analysis (line 210 to line 211). To my professional knowledge, EFA is used to explore the factor structure when it is not known how many factors there are between the items and which factors are determined by which items. In the present study, the researchers just used the scales as a tool to do research. The reviewer fails to see there is any need to perform EFA. Please explain your reason(s).

9- Below the subtitle “Background characteristics of the participants” on page 5, the authors presented Table 1 to describe the sample composition. For the age group, years, the sum of the column % is not equal to 100% (50.8% +35.8%+10.7%+2.6%). Please explain why this could happen.  

10-Age group, years―There is no such saying. The correct expression should be Age (years)

11-In line 219, the authors used p<.05, which is incorrect in format. p should be italicized.

12-Han majority in Table 1 should be changed “Han” and “Other ethnic minorities” should be changed "Minority”.

13-From line 251 to line 265, please replace Cronbach’s a value with the corresponding omega coefficient.

14-In Table 2 on page 7, the authors provided mean score of the scales regarding job stress. However, the description about how these mean values is computed is missing. Please do this work.

15-please remove all characters with “Mean” or “SD” from the 1st column in Table 2. Because they are meaningless at all.

16-In Table 3 on page 8, the authors used logistic regression models to analyze the association between background characteristics and sexual intercourse with non-regular female sex partners and female sex workers in the past six months”. Similarly, in Table 4 on page 9, multiple logistic regression models were used to analyze the associated factors with sexual intercourse with non-regular female sex partners and female sex workers in the past six months. I can see that the logistic regression models that the authors used are linear, however, the authors did not justify this claim. Have you ever attempted non-linear term? If not, why did you claim the relationship between independent variable and dependent variable? Please explain with the investigation of multicollinearity, linearity in the logit, and existence of outliers. Also, please answer this question― What estimation method did you use to estimate the parameters, ML or other estimation method? Furthermore, please explain how you investigated overdispersion in logistic regression issue and identify potential factors that contribute to potential overdispersion (e.g., sampling design, and omitted variables). Moreover, I did not see how the authors assessed the model-data fit (e.g., Hosmer-Lemeshow test statistic, McFadden’s R2, AIC, BIC, and Deviance etc.)

17-The authors did not interpret the meaning of the estimated coefficients. Of course, odds ratios are listed in the Table 3 and Table 4, respectively. However, where is the interpretation? Please use plain English to interpret them.

18-In Table 3 and Table 4, “P values” should be replaced with “p value”.

19-There are many places in which you can find expressions such as “a/b”, which are incorrect.

There are no such saying in English. Please use “either…or”.

*Issues on the Discussion Section

20-Please rewrite this section after re-run your analysis above.

*Other Issues in the Manuscript

21-The reviewer noticed that the decimals that appears in the manuscript with a trailing zero, which is inappropriate in format. Please note that many authorities in scientific, technical, and medical fields recommend that a zero should not be inserted before a decimal fraction when the number cannot be greater than 1 (e.g., correlations, proportions, and levels of statistical significance); that is, “p < 0.05” should be written as “p < .05.”

Round 2

Reviewer 1 Report

All my concern and questions were well answered.

Many thanks to authors. 

Author Response

Thank you for your comments very much. 

Reviewer 2 Report

Reviewer’s Comments

11/18/2022

Thanks for the authors’ revision on the submitted manuscript. It does not get to the bar for publication yet and needs further revision. For ease of communication, the reviewer just listed the issues below for illustration.

 1-The symbol “---” that appears in the title is incorrect. It should be “—".

 2-The abstract is still not that clear and the sentences below could be used to start your writing,

With stratified multi-stage sampling approach, 1361 male factory workers in total in Longhua district of Shenzhen Municipality of China were selected to investigate the multifaced determinants of sexual intercourse with non-regular female sex partners (NRP) and workers (FSW) among them. The results showed that 24.5% and 21.2% participants had sexual intercourse with NRP and FSW in the past 6 months, respectively. More specifically, at individual level, perceived higher level job stress and maladaptive coping styles were linked with higher likelihood of having sexual intercourse with NRP and FSW (add the statistical analysis result here). At interpersonal level, blah blah, and blah (add the corresponding results here).  At social structural level, blah blah, and blah (the same disposal as previously). As for the interaction across levels, blah, blah, and blah, (add statistical analysis results here). After that, theoretical and practical implications of the findings could be briefly stated.

3-This is a scientific research manuscript instead of a narrative. So, please replace all personal pronouns in the texts with passive voice.

 4-The authors should explicitly formulate the research questions instead of the statement of research purposes from line 134 to line 138 on page 3. It is imaginable that a manuscript will be much less attractive to readers if it does not have clear research questions. Meanwhile, it is indeed a question whether readers are willing to continue reading. Although there were many published papers that did not have research questions, I suggest the authors propose research questions with question mark definitively.

5-How do the authors justify the sample size of 1361? The reviewer noticed that there is no description in the manuscript. At the same time, few regression coefficients in Table 3 are statistically significant. One of the possible reasons that result in these issues is the smaller sample size. I suggest the authors refer to the resources at https://statisticseasily.com/2022/05/19/sample-size-logistic-regression/ or at https://www.dartmouth.edu/~eugened/power-samplesize.php for reference.

6-From line 192 to line 240, the authors introduced the one of most important psychometric properties, omega coefficient, of the scales used in the present study. Please use the relevant information from line 299 to line 326 to rewrite the corresponding omega coefficients in the manuscript. As for the texts from line 299 to line 326, please completely remove them.

 7-There are 3 columns in Table 1 on page 6. However, the name of the 1st column is missing. Please use “Characteristics” as the column name.

8-In line 242, the subtitle “2.4 Statistical analysis” could be changed “Statistical analytic strategy”.

 9-From line 242 to line 269, the authors briefly introduced the statistical analytical plan of the present study. Although this paragraph is very informative, the logic is not clear, and readers would be confused with the overwhelming details. The reviewer suggests the authors use the word such as “first”, and “second” to increase the readability of the manuscript. For example, SPSS version 26.0 software were used for statistical analysis. First, descriptive statistics of all categorical variables at each level were computed and presented with mean and standard deviation. Second, item parceling (summing up all items together) and confirmatory factor analysis techniques were used to calculate the value of mean and standard deviation and the reliability coefficient of each subscale, respectively. Third, two logistic regression models were used to explore the determinants of sexual intercourse with non-regular female sex partners (NRP) and workers (FSW) among them. More specifically, briefly describe the procedure from line 257 to line 267.

10-There were no missing values in this study (in line 249). Please move this sentence to an appropriate place below the subtitle “2.2 Participants and data collection” on page 3.

11-The scoring rubric presented in line 197, line 202, line 203, line 210, line 211, line 222, line 223, and line 228 should be added double quotation mark, which should look like this (e.g., from 1 = “never “to 7 = “always”).

12-The character “Brief COPE” appears multiple times in the manuscript, which are inappropriate expressions. It should be written as “Brief COPE scale or inventory). Please correct this issue.

13-In line 217, line 115 and line 116, the authors wrote “social media” and it should be “social media platform popular among Chinese society.

14-The scoring rubric of the scales used in the present study are not on the same metric. For example, the scoring rubric of the perceived social support scale ranges from 0 to 10 while the social media influence scale was scored on a 4-point Likert scale. How do we justify the meaning of the score “1” are the same for both scales? I suggest the authors rescaling the metric to a common scale. You can refer to this resource at https://measuringu.com/convert-point-scales/ or at https://www.ibm.com/support/pages/transforming-different-likert-scales-common-scale

for reference.

15-For the two logistic regression analyses in the study, how did the authors select independent variables? Usually, variables are selected according to professional knowledge and previous studies, and statistically significant association in a univariate analysis.

16-The coding of variables needs to be clarified. In Table 3, there are several places where the value of OR is 1.0, meaning that it is a reference group. Please clarify.

17-The authors provided the results of “OR” and 95% CI of the logistic regression models. However, there are no effect size. Please compute effect size for both logit models. If you do not understand how to address this question, please refers to this resource at https://www.youtube.com/watch?v=W8ktaSKVCL0&t=747s

18-Last but not the least, the authors mentioned and admitted that “the model treats the interaction between factors at each level with equal importance to the influence of factors in a single level” (line 87 to line 88) on page 2. However, there is no interaction term(s) in two logistic regression model. Frankly, it is impossible there is not interaction terms among variables at each level. The present study has so many categorical variables and several continuous variables. Do the authors really confirm that there is no interaction between them? Please add interaction terms in both logistic regression models to check if interaction terms exit.

19-It is good that the authors briefly mentioned the limitations of the study in the 2nd paragraph on page 13. However, one thing is missing—the subscale score created by item parceling technique (summing up the score of the corresponding items of each subscale) is inaccurate. Please google the disadvantages of this technique and compare the use of factor score. I think the authors will understand why the reviewer point out this issue separately.
